# Ultrafast generation and decay of a surface metal

L. Gierster [1,2 ✉], S. Vempati [1,3] & J. Stähler [1,2]

Band bending at semiconductor surfaces induced by chemical doping or electric fields can create metallic surfaces with properties not found in the bulk, such as high electron mobility, magnetism or superconductivity. Optical generation of such metallic surfaces on ultrafast timescales would be appealing for high-speed electronics. Here, we demonstrate the ultrafast generation of a metal at the (10-10) surface of ZnO upon photoexcitation. Compared to hitherto known ultrafast photoinduced semiconductor-to-metal transitions that occur in the bulk of inorganic semiconductors, the metallization of the ZnO surface is launched by 3–4 orders of magnitude lower photon fluxes. Using time- and angle-resolved photoelectron spectroscopy, we show that the phase transition is caused by photoinduced downward surface band bending due to photodepletion of donor-type deep surface defects. The discovered mechanism is in analogy to chemical doping of semiconductor surfaces and presents a general route for controlling surface-confined metallicity on ultrafast timescales.

[1] Fritz-Haber-Institut der Max-Planck-Gesellschaft, Abt. Physikalische Chemie, Berlin, Germany. [2] Humbolt-Universität zu Berlin, Institut für Chemie, Berlin, Germany. [3] Present address: Department of Physics, Indian Institute of Technology Bhilai, Raipur, India. ✉email: gierster@fhi-berlin.mpg.de

When the doping density of shallow donors is increased above a critical value in a semiconductor, the excess electrons, originally localized in hydrogenic potentials at impurity sites, delocalize, and form a metallic band[1]. Remarkably, this Mott or Mott-Anderson transition happens already at low doping densities, for example at parts per $10^4$ atoms in phosphorous-doped silicon[1]. Semiconductor-to-metal transitions (SMTs) also occur in two dimensions at semiconductor surfaces leading to the formation of two-dimensional electron gases (2DEGs)[2]. In the case of oxide surfaces, 2DEG formation is often caused by surface doping with shallow donor defects, such as oxygen vacancies[3,4] or adsorbates as, for instance, hydrogen[5,6]. The positively charged impurity sites modify the surface potential, causing downward surface band bending (BB) of the conduction and valence band (CB and VB, respectively). At low doping density, the BB is concentrated around isolated electron pockets at the surface[6,7]. Electron delocalization occurs above a critical electron density only[2]. 2DEGs have attracted considerable interest in the past years, since, beyond metallicity, they can host phenomena such as magnetism or superconductivity[8,9].

Beyond chemical doping, metal-like properties of semiconductors can also be generated by optical excitation. At low photoexcitation densities the optical and electronic response of semiconductors is dominated by noninteracting free carriers and excitons. In contrast, strong photoexcitation can lead to metal-like behavior by three different mechanisms:

(1) At a critical excitation fluence a Mott transition between free excitons occurs, leading to the formation of an electron-hole plasma with quasi-Fermi levels in the CB and VB[10,11]. This SMT is closely related to the above-described Mott-Anderson transition[12]. In this case, the

material may exhibit metal-like optical properties and conductivities; however, without a change of the equilibrium band structure, there is no density of states around the equilibrium Fermi level $E_F$ and no true SMT has occurred.

(2) A real SMT can be achieved by photoinduced changes to the electronic band structure for instance due to strong carrier-lattice or carrier–carrier interactions that change the screening of the Coulomb interaction[13–15]. Yet, in the case of inorganic semiconductors, strong laser excitation (mJ/cm²) is necessary to drive such photoinduced phase transition (PIPT), as in the famous room temperature PIPT in vanadium dioxide[16–19]. Moreover, due to the high energy uptake, the SMT usually becomes thermally stabilized, and the recovery of the equilibrium phase is limited by thermal dissipation processes of nanosecond duration[20].

(3) Analogous to chemical doping, the band structure can also be optically manipulated by photodoping[21–23]. One pathway is the photoexcitation of deep donors, which creates electron-hole pairs bound in hydrogenic potentials, with the hole localized at the impurity site, see the left hand side of the illustration in Fig. 1c. In contrast to free excitons, these defect excitons are fixed in space, forming a direct analogue to the shallow donors discussed above[24]. As for chemical dopants, the geometric confinement of the photoholes at the surface modifies the surface potential, thereby causing local downward BB. At a critical density of such states, a metallic band is formed (Fig. 1c), leading to surface metallization.

Photoinduced metallization by defect excitons was, so far, only observed on very long timescales in the bulk of semiconductors[25],

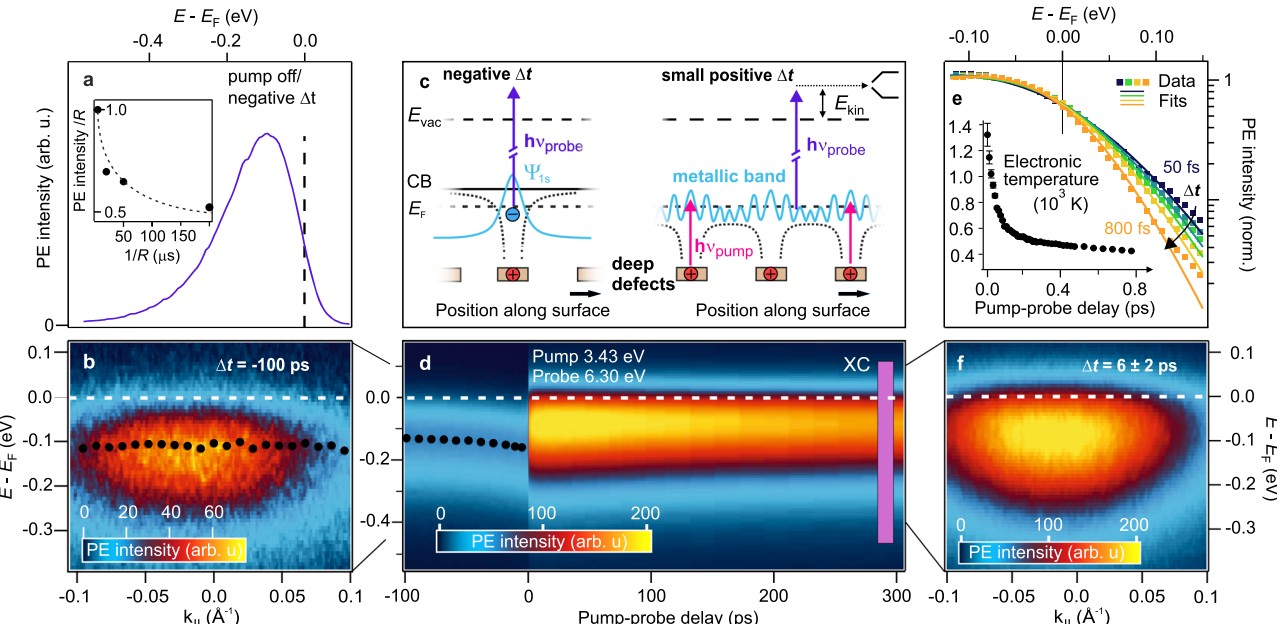

**Fig. 1 Time-resolved photoelectron spectroscopy of the ZnO(10-10) surface upon resonant excitation. a** Angle-integrated photoelectron (PE) spectra recorded with $h\nu_{probe} = 6.3$ eV with the pump laser pulse $h\nu_{pump} = 3.43$ eV off or at negative pump-probe delays $\Delta t$. Inset: Repetition rate dependency of the signal. The dashed line is a guide to the eye. **b** Angle-resolved PE spectrum at negative pump-probe delays showing a dispersionless feature. Black dots: Peak maximum of the intensity distribution. **c** Energy level diagram and illustration of the discussed surface metal generation due to the photoexcitation of deep defects at the ZnO surface. Negative delays: Photodoped semiconductor due to long-lived defect excitons. Small positive delays: Formation of a metallic band. **d** Temporal evolution of the PE intensity in false colors as a function of pump-probe delay $\Delta t$ and energy. Pump laser fluence: 27 μJ/cm². The pump pulse induces an abrupt increase of the electron density below $E_F$. The purple box (XC) indicates the energy-integration window for the evaluation shown in Fig. 2a. **e** Normalized, angle-integrated PE intensity for different positive delays (50–800 fs) with Fermi-Dirac distribution fits (solid lines). Inset: Electronic temperature versus $\Delta t$. Error bars represent standard deviations. **f** Angle-resolved PE intensity averaged from 4 to 8 ps showing a dispersive free electron-like band.

exploiting the effect that the photodoping can be metastable. This leads to persistent photoconductivity, as observed in various semiconductors[25,26]. Among them is the wide band gap (3.4 eV), intrinsically n-doped ($E_F$ ca. 0.2 eV below the CB) semiconductor ZnO, which has a variety of native deep donor defects leading to a broad photoluminescence signal below the fundamental band gap energy[27,28]. For ZnO, a photoinduced, ultrafast control of the conduction properties would be especially appealing, as any application would benefit from the ease of nanostructuring and transparency to visible light of this material[29,30]. At semiconductor surfaces, the photoexcitation of defects would imitate the effect of the gate terminal in field effect transistors. Such phototransistors could then be used for the control of ultrafast currents in information technology or optoelectronic devices, such as light emitters in the teraherz regime[31].

In this article, we unveil an ultrafast photoinduced SMT confined to the surface of ZnO using very low excitation fluences with sub-ns decay and feasible up to at least 256 K. This dramatic effect is enabled by photodoping of the surface: depopulation of deeply bound in-gap (defect) states induces transient local downward BB and populates the CB with electrons. Above a threshold fluence of only 13.6 μJ/cm², the photoexcited electrons delocalize in a non-equilibrium state that shows all defining footprints of a metal: Density of states around the equilibrium Fermi energy $E_F$ resulting from a partially filled dispersive band and an electron distribution following Fermi-Dirac statistics that thermalizes with the lattice within 200 fs. We track the PIPT in the time domain by monitoring the photodoped electron density, the nearly free electron mass of the surface electrons, and the energy shift of the VB. Generation and decay of the surface metal occur on the timescales of electronic screening and electron-hole recombination, respectively. The carrier density of the surface metal shows the same build-up and decay dynamics as the downward shift of the VB that occurs upon positive charging of the surface by photodoping. Moreover, the effective mass evolves as a function of laser fluence with a critical form as expected for the Mott SMT between shallow donor dopants. Remarkably, the ultrafast SMT does not require photoinduced changes to the band structure beyond surface BB; a sufficiently large number of deep donor levels should enable similar effects at the surfaces of many semiconductors.

## Results

**Photostationary n-type doping.** We use time- and angle-resolved photoelectron spectroscopy (trARPES) to monitor the PIPT. A pump-probe scheme, where a first femtosecond laser pulse (pump) excites the sample and a second one (probe) photoemits the non-equilibrium electronic population, allows for the measurement of the ultrafast dynamics. We apply this to a ZnO sample of (10–10) orientation cleaned in ultrahigh vacuum (cf. Methods). Before discussing the ultrafast metallization dynamics, it is important to ascertain the semiconducting initial state. Therefore, we first investigate the sample without applied pump laser pulse. Photoelectron spectra from the VB, similar to those reported in ref. [32], identify the VB maximum at −3.2 eV with respect to $E_F$. This indicates an n-doped sample with the equilibrium Fermi energy 0.2 eV below the CB as is typical for ZnO[28,33]. In order to address the energetic region around $E_F$, we use probe laser pulses with hν = 6.3 eV (energy level diagram in Fig. 1c). This energetic region is interesting, because the electronic structure around $E_F$ defines the conduction properties. For an undoped semiconductor no states in the forbidden gap are expected, while doping with shallow donors below the Mott density induces localized and hence dispersionless states just below $E_F$[6,34]. Figure 1a shows that such states exists at the Fermi level, centered at −0.1 eV below $E_F$ and exhibiting a width of

several hundred meV. An angle-resolved spectrum in Fig. 1b shows that the feature is also dispersionless, indicative of isolated shallow donor dopants.

Beyond chemical doping, shallow dopants could result from photoexcitation of deep defects as outlined in the introduction if their lifetime is sufficiently long to be pumped and probed by two subsequent laser pulses provided by our laser system (200 kHz ≙ 5 μs). As the luminescence of defect excitons in ZnO is known to extend to the μs regime[27], formation of such a photostationary state of long-lived defect excitons would actually be expected. We test this hypothesis by tuning the repetition rate of our laser system from normally 200 kHz down to 5 kHz, which varies the separation of two subsequent laser pulses from 5 to 200 μs. As displayed in the inset of Fig. 1a, the photoelectron intensity drops by almost 50%, showing that a photostationary state is at the origin of the shallow donor signal. The detailed properties of this photostationary state population are out of the scope of the present publication and are discussed elsewhere[35]. We conclude that the dispersionless feature at −0.1 eV below $E_F$ results from photostationary defect excitons that act like shallow donors. We will demonstrate in the following that photoinduced enhancement of the defect exciton and, thus, shallow donor density can lead to metallization of the ZnO surface exclusively on femto- to picosecond timescales after optical excitation.

**The ultrafast PIPT.** In order to drive the ultrafast SMT we excite the sample resonantly with the band gap (hν$_{pump}$ = 3.43 eV) and monitor the photoinduced dynamics with time-delayed (Δt) probe pulses (hν$_{probe}$ = 6.3 eV). With this pump photon energy, it is possible to drive optical transitions across the gap as well as transitions from occupied in-gap states to the CB. Figure 1d shows the evolution of the PE intensity as a function of Δt and energy with respect to $E_F$. The photoresponse of the sample is characterized by a large PE signal below the equilibrium $E_F$, which arises immediately after time zero and persists for several hundred ps. A video of the angle-resolved spectra as a function of Δt gives an impression of the process (Supplementary Movie 1). The angular distribution of the photoelectron intensity, averaged from 4–8 ps, is shown in Fig. 1f. It exhibits a broad feature of few 100 meV width, which is cut by $E_F$ and shows a curvature, to the eye most apparent at low energies. We can extract the curvature by fitting the data with a Gaussian peak multiplied by a Fermi-Dirac distribution (other methods give the same result, see Supplementary Fig. 1). The peak positions describe a parabola with $m_{eff}$ = 1.2(1) $m_e$, i.e., the band nearly exhibits the free electron mass $m_e$. Thus, at small positive delays, ZnO shows all defining characteristics of a metal: a dispersive band, cut by the equilibrium $E_F$, indicative of a SMT.

To further confirm the above finding we analyze the angle-integrated PE spectra in the first ps after excitation (Fig. 1e). The excitation of metals with fs laser pulses initially leads to a heating of the electronic subsystem and a subsequent equilibration with the phonon bath within the first ps[36]. Photoexcited ZnO behaves analogously to photoexcited metals: The spectra exhibit a characteristic change of the high-energy tail around $E_F$, which can be described by a Fermi-Dirac distribution with a delay-dependent temperature (data and fits in Fig. 1e, cf. Methods). The resulting time-dependent electronic temperature is plotted in the inset of Fig. 1e. Starting at 1300 K, it decays due to equilibration with the lattice within approx. 200 fs. The photoinduced phase thus shows electron–electron and electron–phonon interaction as expected for a metal.

To quantify the temporal evolution of the PIPT, we spectrally integrate the photoinduced electron signal below $E_F$ (purple),

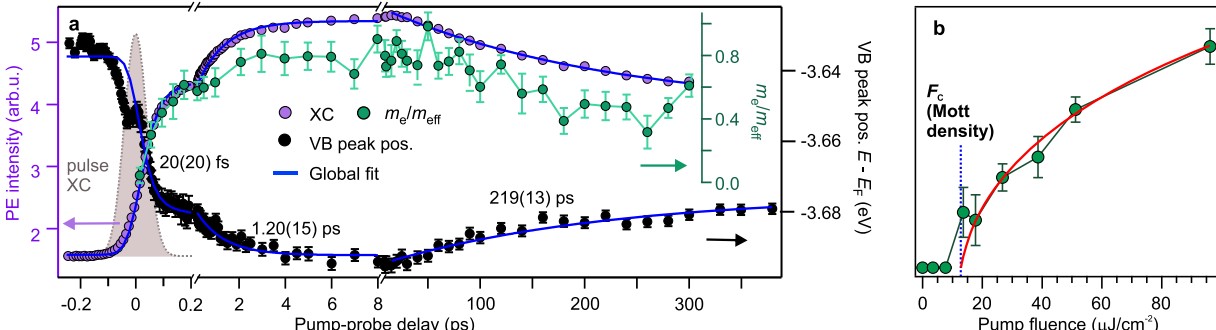

**Fig. 2 Dynamics and fluence dependence of the surface metallization. a** Purple: PE intensity below $E_F$ (proportional to the surface electron density). The energy-integration window is indicated by the purple box (XC) in Fig. 1d. Green: Curvature of the metallic band (inverse effective mass $m_{eff}$, determination see Supplementary Fig. 1). Black: VB peak position (determination see Fig. 3). The blue line is a global fit (see Methods). Gray area: Instrument response function. **b** Fluence-dependent evolution of the band curvature. Red solid line: Fit with the critical form, Eq. (1). Error bars in **a** and **b** respresent standard deviations.

which is proportional to the electron density in the metallic band, and plot its time dependence together with the evolution of the band curvature $1/m_{eff}$ (green markers) in Fig. 2a. Electron density and curvature of the metallic band show a two-fold build-up ($\tau_1 = 20(20)$ fs, $\tau_2 = 1.20(15)$ ps), which is followed by a decay on a timescale of hundreds of ps ($\tau_3 = 219(13)$ ps); see Methods for details on the fitting. As localized states are dispersionless, the rise of the curvature $1/m_{eff}$ can be viewed as dynamic delocalization of the photoexcited electrons and the subsequent decrease of $1/m_{eff}$ during the decay as dynamic localization[37]. The degree of localization is correlated with the photoexcited electron density. We conclude that the free electron-like metal is generated abruptly within the experimental resolution on the timescale of electronic screening[11,38] and decays on a sub-nanosecond timescale.

It should be noted that the photostationary population of shallow donors discussed in the previous section spectrally overlaps with the metallic band. It seems highly likely that the reduction of the surface electron density, accompanied by the dynamic localization, eventually funnels into the localized photostationary population, as discussed later in more detail.

In order to determine the threshold fluence of the PIPT, we tune the excitation fluence across a wide range, from few to tens of μJ/cm². Figure 2b shows the curvature $1/m_{eff}$ as a function of fluence. Below the threshold fluence of $F_c = 13.6$ μJ/cm², the pump laser pulse induces an electron population below $E_F$ with a flat angular distribution (Supplementary Fig. 2a, b), as observed for excitons[38] and also reminiscent of the photostationary response resulting from long-lived defect excitons as discussed above. Also no cooling of a hot thermalized electron population is observed below $F_c$ (Supplementary Fig. 2f, g). Thus, low fluence photoexcitation creates a localized, non-interacting electron population. Above $F_c$, $1/m_{eff}$ increases monotonously and, simultaneously, the hot electron cooling indicative of the metallic phase starts developing (Supplementary Fig. 2h–j). We find $m_{eff} = 0.7\ m_e$ for the highest fluence in our experiment. In order to test whether the effective mass can be viewed as an order parameter of this phase transition, we fit the data in Fig. 2b using the inverse of

$$m_{eff} = A(F - F_c)^{-\alpha} + m_0 \qquad (1)$$

where $A$ is a proportionality constant, $\alpha$ the critical exponent and $m_0$ an offset. The fit coincides with the data. Such critical behavior is predicted from the Mott-Hubbard theory of the Mott SMT[39] and has been observed experimentally upon chemical doping in three[40] as well as in two dimensions[41]. Remarkably, we observe this trend after photoexcitation in the ultrafast time domain,

which suggests that photoexcitation indeed acts like chemical doping already on femtosecond timescales. It should be noted that at high doping density the effective mass of the metallic band would be expected to reach the CB $m_{eff}$[42]. This is confirmed by fixing $F_C$ to 13.6 μJ/cm², resulting in $m_0 = 0.2(6)\ m_e$, which is in agreement with the CB effective mass of 0.25 $m_e$[43].

**The surface photodoping mechanism.** We showed that a photoinduced SMT occurs in ZnO. At low fluence, the excited carriers do not interact beyond exciton formation, which is also in agreement with earlier work on weakly photoexcited ZnO surfaces[44]. Above $F_C$, a transient metal phase is generated. The behavior of the effective mass as a function of pump laser fluence is consistent with a Mott transition, either of free or defect excitons. Such a transition also explains the dynamic change of the effective mass during delocalization (at increasing exciton density) and subsequent localization (at decreasing exciton density).

A crucial aspect of the photoinduced metal phase is that the quasi-$E_F$ in the metallic band equals the equilibrium $E_F$ for all excitation densities (cf. Fig. 1f and Supplementary Fig. 2). This means that the SMT goes beyond a Mott transition between free excitons without changes to the equilibrium electronic band structure: The exciton binding energy in ZnO is 60 meV, which is not enough to create a state at the Fermi level, as the equilibrium $E_F$ is located 200 meV below the bulk CB. Thus, at low fluence, no electronic states below $E_F$ can arise due to free excitons. Likewise, above the Mott density, the free electron-like population in the CB would be 200 meV above $E_F$. Photoexcitation must therefore change the equilibrium band structure. This could be reached, for instance, via band gap renormalization (BGR) due to carrier–carrier screening[10]. BGR would shift the CB downward, and at the same time the VB would move upward[45]. However, also photodoping with defect excitons could lead to electron energies below those expected for free excitons if the doping was confined to the surface: The depopulation of surface defects would result in positive surface charges. This modification of the surface potential leads to local downward BB toward the surface, in analogy to chemical doping of semiconductor surfaces. Such local downward BB can reach several hundred meV below the Mott limit, as shown for ZnO doped with hydrogen[6]. In the case of photodoping, contrary to BGR, the VB should shift downward.

In order to distinguish the BGR and the surface photodoping scenario, we determine the energetic position of the VB upon photoexcitation. A probe photon energy of $h\nu_{probe} = 4.25$ eV gives access to the VB in a two-photon photoemission process (see the inset in Fig. 3a). A pump laser fluence above the SMT threshold $F_C$ is used. Figure 3b compares a VB spectrum at

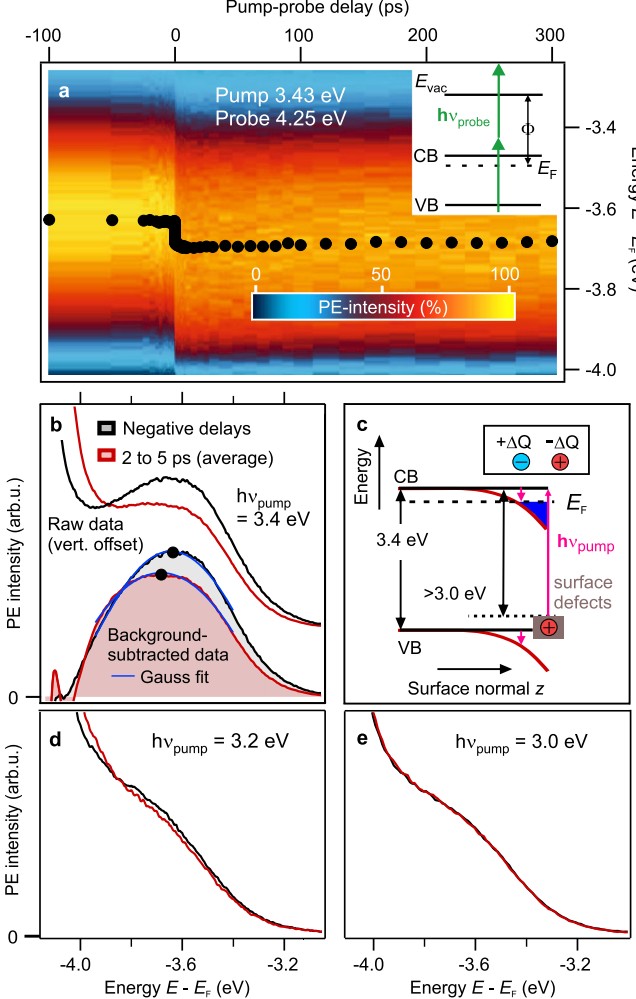

**Fig. 3 Downward shift of the VB upon resonant and below band gap excitation. a** PE intensity from the VB in false colors as a function of pump-probe delay probed with $h\nu_{probe} = 4.25$ eV. This photon energy is below the work function of the sample ($\Phi = 4.4$ eV) and the VB is accessed by two-photon photoemission, see the energy sketch in the inset. The pump photon energy is 3.43 eV at a fluence of 27 μJ/cm². Markers: peak positions, determination see **b**. **b** Comparison of VB spectra at negative delays (black) and at positive delays (red). Top: Raw data (vertically offset for clarity). Bottom: Data after subtracting the background of secondary electrons. Blue: Gaussian fits to determine the peak maximum. **c** Energy level diagram with a sketch of downward surface BB along the surface normal $z$. **d**, **e** VB spectra at negative (black) and positive delays (red) for different pump photon energies below the fundamental gap.

negative delays (black) to one at a positive pump-probe delay of few ps (red). Clearly, the VB is transiently shifted to lower energies. The false color plot in Fig. 3a shows that the downward shift occurs abruptly and persists for several hundred ps. Based on this, we exclude BGR as the driving mechanism and conclude that the depopulation of deep defects at the ZnO surface lies at the origin of the PIPT. It should be noted that BGR most likely still occurs, contributing to the downward shift of the CB. However, the downward shift of the VB shows that the dominant process affecting the electronic band structure is photoinduced surface BB. Note that the VB shift is not entirely rigid but that the peak also appears broadened and has a lower amplitude (Fig. 3b). This observation is consistent with surface BB, where not all probed unit cells along the surface normal exhibit the same shift, as noted previously for chemical doping[46]. Due to this averaging

effect it is impossible to quantify the maximum BB at the very surface[46]. Still, the observed VB position is a qualitative marker of downward BB due to positive surface charging.

Beyond monitoring band positions, trARPES offers another direct way of probing that the surface is indeed positively charged after photoexcitation: As shown previously for GaAs[47], a change of the surface charge leads to a short-ranged electrostatic field that extends into the near-surface vacuum region. In a pump-probe experiment, photoelectrons emitted by the probe laser pulse are decelerated by this pump-induced field at small negative pump-probe delays $\Delta t < 0$, when the photoelectrons are still close to the surface[47]. As discussed in great detail by Yang et al.[47], this effect leads to a downward shift of the energetic position of the detected probe electrons as a function of negative pump-probe delay. The shift is strongest close to time zero and gets weaker at larger negative pump-probe delays[47,48]. In our experiment, the data in Fig. 1d clearly show that such a downward shift of the photostationary state signal probed by $h\nu_{probe} = 6.3$ eV occurs for $\Delta t < 0$ on a 100 ps timescale. The downward shift at negative delays unambiguously demonstrates that the surface is positively charged due to photoexcitation by the pump laser pulse. Note that such photoinduced changes of the surface charge and the resulting change of surface BB are well-known as surface photovoltage phenomena[49]. Both, increase and decrease of BB have been demonstrated in the ultrafast time domain[50].

The final link between the depopulation of deep defects at the ZnO surface and the SMT is given in Fig. 2a, where we compare the VB downward shift as a function of $\Delta t$ (black markers) to the time-dependent electron density in the metallic band (XC, purple markers) obtained under the same excitation conditions. As evidenced by a global fit (blue curve), both transients agree perfectly for all timescales, from ultrafast (abrupt and delayed) rise to decay. We conclude that the surface is photodoped by the photoexcitation of deep defects. Thereby, the pump laser pulse populates the CB with electrons and creates localized positive charges at the surface that induce downward BB (Fig. 3c). At low pump laser fluence, surface-confined defect excitons are formed and the downward band bending is local. Upon crossing $F_C$, a Mott transition occurs and a metallic band at the surface arises (Fig. 1c). Downward BB causes the formation of electronic states in both, the excitonic as well as the surface metal phase, below the equilibrium $E_F$.

**Energetic position and chemical origin of the deep defect levels**. The question about the energetic position and chemical nature of the unexcited deep defect states in the ZnO band gap remains. We address the former by varying the pump photon energy. Photoexcitation will charge the surface positively and lead to the SMT only if the pump photon energy is sufficient to excite electrons from deep donor levels to normally unoccupied states. By tuning the pump photon energy below the fundamental gap of 3.4 eV[28] we can exclusively address in-gap states. As before, the energetic position of the VB in the time domain unveils whether the surface is charged positively by photoexcitation. Figure 3d shows that downward BB is still induced by photoexcitation with $h\nu_{pump} = 3.2$ eV (full width half maximum: 0.1 eV). This unambiguously confirms that, downward BB, and hence the PIPT, is not driven by excitations across the ZnO band gap (3.4 eV). Complementarily, Fig. 3e demonstrates that photoexcitation with $h\nu_{pump} = 3.0$ eV does not induce downward surface BB, i.e., this photon energy is not sufficient to depopulate the relevant defect states. This shows that the in-gap states responsible for the SMT must lie deeply in the band gap, closer than 0.4 eV above the VB maximum. Note that, likely due to the close proximity to the high density of states of the VB maximum, no separate peak due to

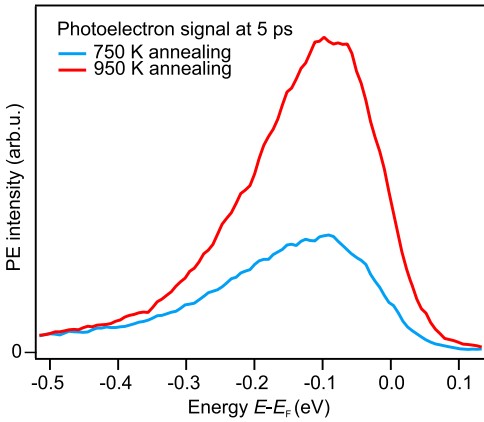

**Fig. 4 Sensitivity of the photoresponse to the surface conditions.** PE spectra recorded with $h\nu_{probe} = 6.3\,eV$ and $h\nu_{pump} = 3.4\,eV$ (fluence $140\,\mu J/cm^2$) at a pump-probe delay of 5 ps using two different annealing temperatures during the sample preparation (cf. methods and see text). The same fluence of pump as well as probe laser pulses is used for both shown spectra, hence the absolute PE intensity can be compared. After annealing at 750 K (blue), the photoelectron intensity of the pump-induced metal phase is by a factor of 2–3 smaller than after annealing at 950 K (red).

the deep defects can be detected in the photoemission data. The experimental spectra only exhibit a strongly broadened VB edge (Fig. 3b).

Defects could be produced at the ZnO surface due to the surface cleaning procedure, which involves annealing in ultrahigh vacuum (cf. methods). The only native defects in ZnO that have a low formation energy and form an occupied state deep in the band gap are lattice vacancies, i.e., oxygen or zinc vacancies[28]. Both can be created during high temperature treatment as shown by mass spectroscopy[51,52]. However, zinc vacancies can be ruled out as they should be negatively charged in thermodynamic equilibrium[28]. Photoexcitation of such states would only diminish the number of negative charges at the surface, and hence reduce existing upward BB instead of creating downward BB. In contrast, oxygen vacancies are neutral in equilibrium and induce a state 0.4 eV above the VB maximum according to hybrid DFT calculations[53], in close agreement with the photon energy dependence reported above.

In order to test the hypothesis that annealing causes the deep defects, we varied the annealing temperature and checked the influence on the photoinduced metallization dynamics. As shown in Fig. 4 the PE intensity of the photoinduced metal phase is higher by a factor of 2–3 when the sample was annealed at 950 K compared to annealing at 750 K. This directly shows that the photoexcitation by the pump laser pulse addresses states created during the annealing step at the ZnO surface.

The above observation compares well with previous work by ref. [51], who found that oxygen vacancies are formed above 700 K with increasing efficiency as the temperature is raised. Hence, it seems likely that oxygen vacancies cause the deep defect levels.

## Discussion

The photodoping process and the associated timescales are illustrated in Fig. 5. Upon arrival, the pump laser pulse depopulates deep donor defects at the surface and populates the CB with electrons (process 1). Subsequently, with $\tau_1 = 20(20)\,fs$, downward surface BB builds up, shifting the CB below $E_F$. In the regime of low fluence photoexcitation by the pump laser pulse, localized electron pockets are formed in proximity of the surface,

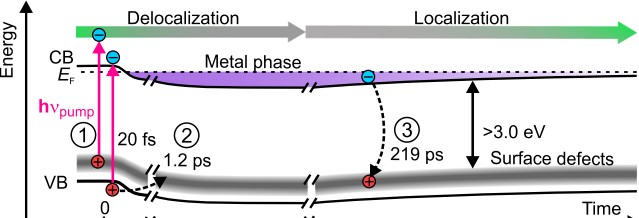

**Fig. 5 Scheme of the pump-induced processes.** Photoexcitation resonant to the ZnO band gap populates the CB and depopulates surface defect states leading to downward surface BB; above the threshold fluence $F_C$ the electrons delocalize and form the metal phase (process 1). Delayed increase of the BB occurs due to hole trapping at surface defect sites (process 2). The PIPT recovers as the surface charge is reduced by electron-hole recombination (process 3).

and the BB is a local effect only[6]. An electron bound to a charged impurity site is part of a defect exciton. This is conceptually identical to a shallow donor dopant. Above the critical photo-excitation fluence $F_c = 13.6\,\mu J/cm^2$, a Mott transition occurs: The CB electrons, originally localized at the defect centers, delocalize and generate the surface metal phase. This delocalization (decreasing $m_{eff}$) occurs simultaneously with the band structure change on a timescale of electronic screening.

After the abrupt laser excitation, the downward shift of the VB, the electron density of the surface metal, and the degree of delocalization increase further with $\tau_2 = 1.20(15)\,ps$. We interpret this as hole trapping at surface defect sites (process 2), which enhances the photodoping density at the surface. Consequently, the surface is charged even more positively, which leads to an increase in surface BB. A resolution-limited upper boundary of 80 ps for this process was recently identified by X-ray absorption spectroscopy[54] and few-ps time constants were determined by optical spectroscopy[27]. However, we note that hole polaron formation[55] may result in similar dynamics.

Eventually, part of the CB electrons recombine with the surface defects and, as the doping density is reduced, the remaining electrons localize (increasing $m_{eff}$) with $\tau_3 = 219(13)\,ps$ (process 3). This back-transition to the semiconducting state is at least one order of magnitude faster than, for example, the decay of the photoinduced SMT in VO$_2$[20]. A fraction of the remaining defect excitons has a lifetime that exceeds the inverse repetition rate of our laser system. They lead to the photostationary n-type doping of the surface and appear as the shallow donor signal at negative delays.

Interestingly, the above processes do not require cryogenic temperatures: The PIPT can be realized between 100 K up to at least 256 K, i.e., close to room temperature (Supplementary Fig. 3).

In summary, we unveiled a photoinduced ultrafast SMT at the surface of ZnO with a very low threshold fluence. The mechanism is simple and universal: Photodepletion of in-gap states (deep donors) causes photodoping of the surface, which leads to local downward BB and eventually a partially filled metallic band below $E_F$. All our experimental observations are consistent with this mechanism, from positive surface charging and downward BB that occurs simultaneously with the SMT, to the (dynamic) delocalization of the surface electrons. The same mechanism should lead to PIPTs at surfaces or interfaces of other semiconductors with a sufficiently high density of donor-type deep surface defects. Beyond technological implementations of ZnO as an ultrafast, transparent photoswitch with the transient properties of an equilibrium metal, this work is the starting point for studies of 2DEGs with emerging properties beyond metallicity in the ultrafast time domain.

## Methods

**Sample preparation**. The ZnO(10–10) sample was purchased from MaTeck GmbH and is prepared by repeated cycles of $Ar^+$-sputtering (0.75 keV, 8 μA) succeeded by 30 min annealing in ultrahigh vacuum ($T_{max}$ = 950 K, heating rate 30–40 K min$^{-1}$). The surface cleanliness is confirmed by LEED and photoemission measurements of the work function.

**trARPES measurements**. Photoemission measurements are performed in situ using a hemispherical electron energy analyzer (PHOIBOS 100, Specs GmbH). A bias voltage of 1.5 eV is applied with respect to the sample. The $E_F$ reference is taken from the gold sample holder, which is in electrical contact with the sample surface. Pump and probe laser pulses are generated from a regenerative amplifier system running at 200 kHz and several optical parametric amplifiers (OPAs) working at 200 kHz (PHAROS, internal OPA, Orpheus-N-2H, Orpheus-N-3H by Light Conversion). 6.3 eV laser pulses are reached by frequency quadrupling of the 1.55 eV output of the internal OPA. Pump pulses with 3.2–3.4 eV and probe pulses with 4.25 eV are created in the Orpheus-N-2H and 3H, respectively. All measurements shown in the main text are performed at 100 K.

**Fit functions**. The fit functions for extracting the time constants (Fig. 2a) consist of a double exponential rise and a single exponential decay convolved with a Gaussian peak representing the cross correlation of pump and probe laser pulses. The latter is determined from high-energy cuts in the ZnO pump-probe data in case of using 6.3 eV as probe photon energy, yielding a cross correlation width of 115(17) fs (full width half maximum). In the case of using 4.25 eV as probe photon energy, the cross correlation is measured in situ by pump-probe photoemission from a tantalum sheet yielding 71(1) fs (full width half maximum). The rise and decay time constants are the result of a global fit to the PE intensity around $E_F$ and to the VB shift, as shown in Fig. 2a. The time resolution in the experiments is limited by the accuracy of determining the pump-probe cross correlation that is used for the convolution.

The fit function to extract the electronic temperature of the metallic state is a Gaussian peak multiplied by a Fermi-Dirac distribution, convolved with a Gaussian peak accounting for the energy resolution of the experiment (50 meV).

## Data availability

A minimal dataset (source data for Figs. 1b, d, e, f, 2a and 3a, b) to reproduce the main findings of the present study is deposited in the Zenodo repository with the identifier https://doi.org/10.5281/zenodo.4421027. All other data analysed during the current study is available from the corresponding author on reasonable request.

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

## Acknowledgements

This work was funded by the Deutsche Forschungsgemeinschaft (DFG, German Research Foundation)—Project-ID 182087777—SFB 951. S.V. would like to thank the Max Planck Research Society for a Max Planck Postdoctoral Fellowship.

## Author contributions

L.G. and S.V. did the experiments, L.G. analyzed the data. L.G. and J.S. wrote the manuscript. J.S. guided the work. All authors contributed to discussions.

## Funding

## Competing interests

The authors declare no competing interests.
