## [Peer Review File · Nature Communications]

REVIEWER COMMENTS

Reviewer #1 (Remarks to the Author):

The authors present an experimental study of the semiconductor-to-metal transition at the surface of ZnO. They use time-resolved ARPES to track the dynamic band structure, and observe a dramatic change in the low energy density of states upon the excitation with very low laser fluence. The results are explained by the photoinduced surface band bending, involving the in-gap defect states.

The general subject of the laser-induced phase transition in materials is timely and significant, for not only understanding the nonequilibrium physics but also for potential ultrafast applications. I believe that this work represents a new contribution to this field with interesting findings, the experimental data is of high quality and the manuscript is well written. However, I have concerns about the interpretation of the experimental results and the discussion of the mechanism, which I think should be further clarified. My overall recommendation is that the manuscript shall not be published in Nature Communications in the present form, but I would be very happy to see a revised version. Please see following my detailed comments/suggestions.

1) Since the surface defect states play a central role in the proposed surface band bending scenario, one would expect to see more evidences about such states and the related photo depletion in the paper. It would be necessary to show the surface defect states in the experimental band structure, not only the cartoon in Fig. 4.

2) The results in Fig. 3 show that the electronic response is large only with the pump resonant to the band gap. In this case, it seems that the transition from the valence band to conduction band, rather than the photo depletion of the defect states, is dominant in the initial photoexcitation process. To clarify this point, it would be helpful to check the intensity dynamics of both the valence band and defect states.

3) The direct optical transition from the defect states to conduction band, if it is indeed the dominant process, should only happen near the defects in the real space. Then it takes time for the electrons to delocalize and form the long range order and dispersive band, and this timescale should depend on the defect density and laser fluence. It would be interesting to make a rough estimation based on the observed timescale of ~ 20 fs, and check if the defect density is reasonable or not for the sample used in the experiment.

4) In the usual band bending picture, one would expect to see a nearly rigid band shift, as illustrated in Fig. 3c and Fig. 4. The observed shift of the valence band is about 60 meV in Fig. 2a. However, the conduction band has to shift for more than 200 meV to touch the Fermi level, if the observed low energy state after laser pump (Fig. 1c) is the conduction band. In this sense, one may also need to discuss about the band gap renormalization.

5) What is the band/state located below the Fermi level (the black dots in Fig. 1b) before laser excitation? Is that the in-gap defect state? Is there any possibility that this band/state is the same one observed after laser excitation? It would be helpful to show the ARPES spectrum before laser excitation.

6) As the authors mentioned in Line 189, "a fraction of the photodoped and localized surface electrons show lifetimes exceeding the inverse repetition rate of our laser system (5 μ s)", the laser-excited material does not fully recover before the arrival of next pulse, which is harmful for the pump-probe measurement. The reliability of the experimental data should be justified in presence of such an

accumulation effect.

Reviewer #2 (Remarks to the Author):

This work has demonstrated the observation of ultra-fast semiconductor-to-metal (SMT) in crystalline ZnO material with a relatively low threshold fluence compared to other reports. The time/angular-resolved photoelectron microscopy (trARPES) are mainly utilized to elucidate the underlying mechanisms where the downward band bending of valence band induced by the positively charged deep states are attributed to the SMT.

According to this study, the metal-like behavior is supported by the observed dispersive band, nearly-free electron mass, and the observation of time-dependent electronic temperature. Furthermore, the band bending mechanism proposed by the authors is evident by analyzing the results of trARPES with different pumping energies, where the bandgap renormalization is subsequently excluded.

The defect-assisted and photo-induced phase transition is indeed an interesting and practical aspect to the community, as the defects states are inevitable in any kind of material system. The reviewer's comments can be found as follows:

1. It is claimed in the paper that the photoinduced SMT is confined at the ZnO surface instead from the bulk region. In addition, the reviewer notice that the ZnO sample was subjected to the Ar-sputtering process with high-temperature annealing before any measurement. Is this preparation step related to the formation of surface metal? Since the observed SMT is attributed to the deep defect states, the sputtering process might unintentionally introduce defect states in the ZnO surface region. If so, can the transition speed be further improved by engineering the surface defects in ZnO?
2. A probing energy of 4.25 eV is selected to access the VB information in the measurement. How did the authors decide to use this value instead of others? Could the authors provide a brief discussion in this aspect?
3. Could the authors clarify more in details for the argument mentioned in pp. 5, line 150-153? Why the positive surface charges will decelerate the photoelectrons only at the small negative pump-probe delay?
4. It is evident that the deep defect states are responsible for this transition process; however, the origin of these defects are not mentioned in the manuscript. It will be better if the authors can discuss more about the physical origin of the deep defect states. This could help the readers find possible engineering methods to demonstrate similar phenomenon in other types of transition metal oxide (TMO) materials.
5. In the beginning of this manuscript, it is found that there is lack of discussion in terms of the potential applications for this ultra-fast and low fluence SMT in ZnO.
6. In line 154, the electron signal was downward shifted in the negative delay region. It was explained as a photoelectron deacceleration by the pump-induced positive surface charges. What do the black dots in Fig.1(a) stand for? And why there is no similar shift in negative delay region in VB as shown in Fig.3(a)?
7. In line 167 the author introduced a method to address the in-gap states. It was achieved via tuning

the pump energy ($h\nu_{\text{pump}}=3.0/3.2$ eV) and recording the threshold energy that triggered the PE intensity shift. However, there was no evidence that the photon excitation was purely resulted from deep-level trap delocalization.

8. In line 192 the influence of temperature was mentioned (100K-256K). It was depicted in Extended Data Fig. 3 that higher PE intensity was observed at higher temperature. What I observe is that at 256K, the equilibrium PE intensity is almost the same as the excited one ($\Delta t=50\text{fs}$) at 97K. Does it imply that the PE intensity variation could also be a result of heating by the laser?

9. Overall, the manuscript provides a detailed description on the PIPT method to monitor the energy states at the ZnO surface. It claims that the SMT after photoexcitation is driven by BB. The method is interesting to me, but the result is "not surprising". In the view of providing stronger evidence and enriching the manuscript, I suggest adding more probing methods or compare the PIPT results with other methods. For instance, DLTS can be used as a supporting data to the argument of the deep level traps.

10. Most of the fonts are not very clear. Please improve the graph.

Reviewer #3 (Remarks to the Author):

The manuscript by L. Gierster et al. reports on the experimental observation of the ultrafast generation and decay of a metallic surface state in ZnO upon weak photoexcitation. The reported results are without doubt novel and very interesting and envision potential future applications particularly with respect to the applied fluence range as well as the observed characteristic (ultrashort) time scales. The presented trARPES data are of very high quality and in general support the interpretation of the authors (see, however, my comment below). Particularly the formation of a (real) transient metallic state on extremely short timescales is convincingly proven by the experiment. Based on the topic, the novelty of the presented results and the convincing data the manuscript seems to me very suitable for publication in Nature Communications. However, before a final decision I would like to ask the authors to consider the following comments:

The authors argue that the formation of the metallic state is result of a (rigid?) shift of the conduction band below the Fermi level upon photoexcitation. This is implied by Figure 1(a) and also suggested as the favorable scenario later in the manuscript. However, in the transient data [Figure 1(b) and also the attached movie] I cannot see indications for such a transient band shift (a shift of the peak maximum associated with the band) neither during the formation nor during the decay of the metallic state. Instead, the authors convincingly show that the effective mass of the band is changing as a function of time, excluding at least a rigid(!) shift of the conduction band. As an alternative I am wondering, whether (also) the following scenario could apply: As the excited carrier density is increased during the absorption of the pump pulse, first, an excitonic state right at the Fermi level is formed that then transiently undergoes an exciton Mott-transition. Such a scenario could explain the change in the effective mass as a function of time and seems also to be supported by the fluence dependent data shown in Figure 2. I would like to emphasize that such an interpretation would not contradict the key message of the paper. However, it would rather fit to the first scenario listed in the introductory part of the manuscript.

In Fig.1(b) before time-zero a spectral feature of minimum PE intensity is marked which shows a transient behavior due to the interaction with the positively charged surface (this detail is discussed in the manuscript). Can the authors comment on the origin of this spectral feature: In the movie one recognizes also for this feature a parabolic dispersion out of which the metallic band is kind of emerging.

Reply to Referee 1

The authors present an experimental study of the semiconductor-to-metal transition at the surface of ZnO. They use time-resolved ARPES to track the dynamic band structure, and observe a dramatic change in the low energy density of states upon the excitation with very low laser fluence. The results are explained by the photoinduced surface band bending, involving the in-gap defect states.

The general subject of the laser-induced phase transition in materials is timely and significant, for not only understanding the nonequilibrium physics but also for potential ultrafast applications. I believe that this work represent a new contribution to this field with interesting findings, the experimental data is of high quality and the manuscript is well written. However, I have concerns about the interpretation of the experimental results and the discussion of the mechanism, which I think should be further clarified. My overall recommendation is that the manuscript shall not be published in Nature Communications in the present form, but I would be very happy to see a revised version. Please see following my detailed comments/suggestions.

We thank the referee for his/her positive assessment and his/her comments/suggestions, which were useful and helped improving the manuscript. We address them in the following.

1) Since the surface defect states play a central role in the proposed surface band bending scenario, one would expect to see more evidences about such states and the related photo depletion in the paper. It would be necessary to show the surface defect states in the experimental band structure, not only the cartoon in Fig. 4.

In the experimental band structure, these defect states cannot be clearly distinguished. This is because the energetic position of the surface defects in the ZnO band gap are closer than 0.4 eV to the VB maximum according to the pump photon energy dependence (Fig. 3). The valence band itself has a very high density of states (hence large photoelectron signal) and is strongly broadened such that it overlaps with the signal due to the defect states. Photoelectron spectra of this energy region are shown in Fig. 3b.

We agree with the referee that since surface defect states play a central role in the discussed mechanism for the photoinduced semiconductor-to-metal transition at the ZnO surface, more evidence of such states would be desirable. In order to substantiate that the transition from across the fundamental gap cannot be responsible for the PIPT, we included additional data in the

manuscript, showing that photoexcitation by the pump laser pulse must address states that are created during high temperature annealing in the surface preparation process.

Related changes in the manuscript:

- Added Line 255: “Note that, likely due to the close proximity to the high density of states of the VB maximum, no separate peak due to the deep defects can be detected in the photoemission data. The experimental spectra only exhibit a strongly broadened VB edge (Fig. 3b).”
- Added Fig. 4
- Added Line 259: “Defects could be produced at the ZnO surface due to the surface cleaning procedure, which involves annealing in ultrahigh vacuum (cf. methods). [...]

In order to test the hypothesis that annealing causes the deep defects, we varied the annealing temperature and checked the influence on the photoinduced metallization dynamics. As shown in Fig. 4 the PE-intensity of the photoinduced metal phase is higher by a factor of 2-3 when the sample was annealed at 950 K compared to annealing at 750 K. This directly shows that the photoexcitation by the pump addresses states created during the annealing step at the ZnO surface.”

2) The results in Fig. 3 show that the electronic response is large only with the pump resonant to the band gap. In this case, it seems that the transition from the valence band to conduction band, rather than the photo depletion of the defect states, is dominant in the initial photoexcitation process. To clarify this point, it would be helpful to check the intensity dynamics of both the valence band and defect states.

The electronic response, i.e. the valence band downward shift in Fig. 3, starts with a pump photon energy of 3.2 eV. It is because of this that we exclude photoexcitation across the band gap (3.4 eV) as the cause of the downward shift. The photoresponse is, however, indeed stronger with a photon energy of 3.4 eV as pointed out by the referee. Figure R1 shows why a larger photoresponse is expected when the pump is 3.4 eV *without* involving photoexcitation across the fundamental gap. The in-gap states due to the surface defects are localized in real space and hence broad in reciprocal space. With 3.2 eV only transitions at the center of the Brillouin zone are possible (Fig. R1 a). Upon using higher photon energies, transitions to the CB at finite wave vectors can be excited (Fig. R1 b). This shows that the photoexcitation cross section is expected to be larger with 3.4 eV than with 3.2 eV.

Figure R1 Electronic band structure of ZnO(10-10) with in-gap states close to the VB. a) 3.2 eV is just enough to excite an electron in the CB bottom at the center of the Brillouin zone. b) With 3.4 eV, transition at finite k vectors become possible, which enhances the photoexcitation cross section. Note the bottom axis is momentum parallel to the surface. The VB and CB actually are paraboloids in the three dimensional k -space. As commonly done the displayed band structure is projected along k_z ,

The observed photon energy dependence is also in agreement with other experimental studies. As we discuss in the revised manuscript (lines 259-276), the most likely defect is the oxygen vacancy, created by high temperature annealing. The absorption of such defects is known to be indistinguishable from the fundamental absorption edge in optical spectroscopy [Wang, J. et al. ACS Appl. Mater. Interfaces **4**, 4024–4030 (2012), <https://doi.org/10.1021/am300835p>]. Also, the photoluminescence from oxygen vacancies is most intense after photoexcitation with 3.4 eV [Leiter, F. et al. Physica B **340-342** 201-204 (2003) DOI:10.1016/j.physb.2003.09.031], showing that they can be most efficiently excited with this photon energy, in agreement with our experiment.

It would certainly be nice to measure the intensity dynamics of valence band and the defects states. However, addressing them this is impossible due to the spectral overlap of these states in the photoelectron spectra (cf. our answer to question 1)) as well as due to the large spectral changes that occur in this energetic region as the VB is transiently shifted downward (cf. our answer to question 4)).

3) The direct optical transition from the defect states to conduction band, if it is indeed the dominant process, should only happen near the defects in the real space. Then it takes time for the electrons to delocalize and form the long range order and dispersive band, and this timescale should depend on the defect density and laser fluence. It would be interesting to make a rough estimation based on the observed timescale of ~ 20 fs, and check if the defect density is reasonable or not for the sample used in the experiment.

We estimate the delocalization time from the plasma frequency of the electronic system, because this is the time scale at which electronic correlations evolve [Huber, R. *et al. Nature* **414**, 286–289 (2001), DOI: 10.1038/35104522]. The plasma frequency is given by

$$\omega = \sqrt{\frac{e^2 n_e}{m_{\text{eff}} \varepsilon}}$$

where n_e is the electron density, m_{eff} the effective mass, and ε the dielectric constant. The electron density in the metallic state at the surface can be estimated from the radius of the Fermi surface, assuming that it is a circular Fermi surface with the radius given by the Fermi cut through k_F of the surface metal via [Ozawa, K. & Mase, K. *Phys. Rev. B* **81**, 205322 (2010).]

$$n_{2D} = k_F^2 / 2\pi.$$

We extract k_F from the measured band structure for an exemplary fluence of $26 \mu\text{J}/\text{cm}^2$, which is the pump laser fluence for which we show the temporal evolution of the electron density and degree of delocalization (effective mass) in Fig. 2a. The electron density in the surface metal and the degree of delocalization evolve concomitantly, but for a coarse estimate we take the band structure at 6 ± 2 ps (Extended Data Fig. 1a). Taking the dispersion extracted from Fermi-Dirac Gauss fits (blue dots in Extended Data Fig. 1a), k_F is determined to be 0.1 \AA^{-1} . Hence n_{2D} is $1.5 \times 10^{13} \text{ cm}^{-2}$, which is close to what is found in the 2DEG due to hydrogen adsorption at ZnO(10-10) by ref. [Ozawa, K. & Mase, K. *Phys. Rev. B* **81**, 205322 (2010)].

Local band bending at ZnO(10-10) extends over about one nm only [Deinert, J.-C., *et al Phys. Rev. B* **91**, 235313 (2015).] for H-doping. Confining the charge density of the metallic band in our experiments analogously to 1 nm, the electron density is $n_{3D} = 1.5 \times 10^{20} \text{ cm}^{-3}$. The plasma frequency can be finally estimated to $2 \times 10^{14} \text{ Hz}$ (taking $\varepsilon = 8 \varepsilon_0$) for ZnO), i.e. the plasma period would be about 30 fs.

This estimation is clearly very coarse, but illustrates that the delocalization can indeed be very fast and that the extracted delocalization time of 20(20) fs is reasonable.

It should be noted that the estimated electron density $n_{2D} = 1.5 \times 10^{13} \text{ cm}^{-2}$ is on the order of the estimated density of oxygen vacancies at vacuum-annealed (and UV illuminated) ZnO(10-10), which is in the range of 10^{12} - 10^{13} cm^{-2} [Göpel, W. & Lampe, U. *Phys. Rev. B* **22**, 12 (1980)]. As mentioned above, in the revised manuscript we show that the surface defects responsible for the photoinduced metallization are created upon vacuum annealing and are most likely oxygen vacancies (Fig. 4 and lines 259-276 in the revised manuscript).

4) In the usual band bending picture, one would expect to see a nearly rigid band shift, as illustrated in Fig. 3c and Fig. 4. The observed shift of the valence band is about 60 meV in Fig. 2a. However, the conduction band has to shift for more than 200 meV to touch the Fermi level, if the observed low energy state after laser pump (Fig. 1c) is the conduction band. In this sense, one may also need to discuss about the band gap renormalization.

We agree with the referee that the valence band shift should be discussed in further detail. It cannot be compared *quantitatively* to the conduction band downward shift, because the photoemission measurement averages over several unit cells with different amount of band bending. Band bending is confined to about 1 nm to the surface and, furthermore, it is strongest around the charged defect sites. This has been shown by DFT calculations for hydrogen-doped ZnO surfaces [Deinert, J.-C., *et al* Phys. Rev. B **91**, 235313 (2015)]. The valence band photoemission measurement averages, firstly, along the surface normal because of the finite probing depth (of several nm typically, determined by the mean free path of photoelectrons) and, secondly, across the surface.

Since we observe positive surface charging, and show that the surface metal phase is intimately connected to this (concomitant evolution of VB downward shift and electron intensity in the metallic band as a function of pump-probe delay in Fig. 2a), we exclude band gap renormalization as the driving mechanism for the photoinduced phase transition. Yet, we agree that a contribution of BGR to the overall VB shift cannot be excluded and discuss this now in the manuscript.

Related changes in the manuscript

- We wrote in the original manuscript

Line 211: “Note that the VB shift is not entirely rigid but that the peak also appears broadened and has a lower amplitude (Fig. 3b). This observation is consistent with surface BB, where not all probed unit cells along the surface normal exhibit the same shift, as noted previously for chemical doping⁴⁷.”

In the revised manuscript we added

Line 214: “Due this averaging effect it is impossible to quantify the maximum BB at the very surface⁴⁷. Still, the observed VB position is a qualitative marker of downward BB due to positive surface charging.”

- Added

Line 209: “It should be noted that BGR most likely still occurs, contributing to the downward shift of the CB. However, the downward shift of the VB shows that the dominant process affecting the electronic band structure is photoinduced surface BB.”

5) What is the band/state located below the Fermi level (the black dots in Fig. 1b) before laser excitation? Is that the in-gap defect state? Is there any possibility that this band/state is the same one observed after laser excitation? It would be helpful to show the ARPES spectrum before laser excitation.

We agree that it is helpful to discuss the signal at negative pump-probe delays/before laser excitation with the pump pulse in more detail. We now incorporate ARPES spectra observed at negative delays and a discussion in the manuscript. Briefly, the signal at negative delays is attributed to long-lived defect excitons. The long-lived state is not the deep defect state but is due to the *photoexcited* deep defects. Because the species is so long lived, it forms a photostationary state, as we suggested in the original manuscript:

Line 189: “a fraction of the photodoped and localized surface electrons show lifetimes exceeding the inverse repetition rate of our laser system (5 μ s)”

We believe that the state just after photoexcitation with the pump laser pulse (i.e. at positive delays) is the same. In the revised manuscript we now discuss this explicitly and in great detail in an own section (“Photostationary n-type doping”). The pump laser pulse photodepletes deep defects. At low pump fluence, defect excitons are formed. At a critical density they undergo a Mott transition to the metallic phase. The metallic phase is restricted to picosecond timescales after optical excitation, where a sufficiently high density is reached. After reduction of the defect exciton density in the ultrafast time domain by recombination, a long-lived fraction of these states remains.

Related changes in the manuscript

- Added two panels to Figure 1 with photoelectron spectra (angle-integrated and angle-resolved) at negative delays/with the pump laser pulses off.
- Added Line 89 to 118: Section “Photostationary n-type doping.”
- Added Line 155: “It should be noted that the photostationary population of shallow donors discussed in the previous section spectrally overlaps with the metallic band. It seems highly likely that the reduction of the surface electron density, accompanied by the dynamic localization, eventually funnels into the localized photostationary population, as discussed later in more detail. “

6) As the authors mentioned in Line 189, “a fraction of the photodoped and localized surface electrons show lifetimes exceeding the inverse repetition rate of our laser system ($5 \mu\text{s}$)”, the laser-excited material does not fully recover before the arrival of next pulse, which is harmful for the pump-probe measurement. The reliability of the experimental data should be justified in presence of such an accumulation effect.

Indeed, long-lived photoexcited states lead to an accumulation effect if the lifetime exceeds the inverse laser repetition rate. However, after a build-up time a photostationary state is reached which results from the repetitive population *and* depopulation of this state. This scenario is illustrated in Figure R2.

Our data shows that this scenario applies. By lowering the repetition rate, we observe a reduction of the intensity of the long-lived state (cf. the inset in Fig. 1a in the revised manuscript). This is because the photoexcited population has more time to decay, thereby lowering the photostationary population. Upon increasing the repetition rate again, the original intensity is resumed. The corresponding scans are shown in Figure R2 b) for convenience of the referee. The photostationary state is reached within less than 50 ms (the minimum time it takes in our setup to take a photoelectron spectrum) and then remains stable.

Figure R2 **a**, Accumulation effect leading to a photostationary state after repetitive laser excitation. **b**, Photoelectron intensity of the long-lived defect exciton as a function of time, probed with $h\nu_{\text{probe}}=6.3 \text{ eV}$. The laser repetition rate was tuned from 200 kHz down to 5 kHz, as indicated in the figure legend, while illuminating the same spot at the sample surface. Lowering the repetition rate lowers the PE intensity of the photostationary population (cf. the inset in Fig. 1c in the revised manuscript). Upon switching back to 200 kHz the initial intensity is recovered. Note that the time window of 600 seconds corresponds to several million laser excitation events. The fact that the photoelectron intensity is stable over such many excitation events means that a photostationary equilibrium is reached (indicated as n_{∞} in **a**).

Reply to Referee 2

This work has demonstrated the observation of ultra-fast semiconductor-to-metal (SMT) in crystalline ZnO material with a relatively low threshold fluence compared to other reports. The time/angular-resolved photoelectron microscopy (trARPES) are mainly utilized to elucidate the underlying mechanisms where the downward band bending of valence band induced by the positively charged deep states are attributed to the SMT.

According to this study, the metal-like behavior is supported by the observed dispersive band, nearly-free electron mass, and the observation of time-dependent electronic temperature. Furthermore, the band bending mechanism proposed by the authors is evident by analyzing the results of trARPES with different pumping energies, where the bandgap renormalization is subsequently excluded.

The defect-assisted and photo-induced phase transition is indeed an interesting and practical aspect to the community, as the defects states are inevitable in any kind of material system. The reviewer's comments can be found as follows:

1. It is claimed in the paper that the photoinduced SMT is confined at the ZnO surface instead from the bulk region. In addition, the reviewer notice that the ZnO sample was subjected to the Ar-sputtering process with high-temperature annealing before any measurement. Is this preparation step related to the formation of surface metal? Since the observed SMT is attributed to the deep defect states, the sputtering process might unintentionally introduce defect states in the ZnO surface region. If so, can the transition speed be further improved by engineering the surface defects in ZnO?

In the original manuscript we mentioned that the deep defect states “can be caused by lattice vacancies³⁰, which are created at the surface upon annealing in ultra-high vacuum⁴⁵.” (line 164-165 of the original manuscript). In the revised manuscript we added additional data that evidences that the deep defect states that are photoexcited and cause the PIPT must indeed be created during the sputtering/annealing process and are most likely oxygen vacancies. These defects are located at the surface, hence the photoinduced downward band bending, and the associated surface confinement of the photoinduced SMT.

Regarding the question if the transition time can be further enhanced due to engineering the surface defects we cannot give a definite answer with our experiment. The transition time of 20(20) fs is already at the limit of our experimental resolution.

Related changes in the manuscript:

- Added Fig. 4
- Added Line 259: “Defects could be produced at the ZnO surface due to the surface cleaning procedure, which involves annealing in ultrahigh vacuum (cf. methods). The only native defects ZnO that have a low formation energy and form an occupied state deep in the band gap are lattice vacancies, i.e. oxygen or zinc vacancies²⁸. Both can be created during high temperature treatment as shown by mass spectroscopy^{51,52}. However, zinc vacancies can be ruled out as they should be negatively charged in thermodynamic equilibrium²⁸. Photoexcitation of such states would therefore only diminish the number of negative charges at the surface, and hence *reduce existing* upward BB instead of *creating* downward BB. In contrast, oxygen vacancies are neutral in the equilibrium and induce a state 0.4 eV above the VB maximum according to hybrid DFT calculations⁵³, in close agreement with the photon energy dependence reported above.

In order to test the hypothesis that annealing causes the deep defects, we varied the annealing temperature and checked the influence on the photoinduced metallization dynamics. As shown in Fig. 4 the PE-intensity of the photoinduced metal phase is higher by a factor of 2-3 when the sample was annealed at 950 K compared to annealing at 750 K. This directly shows that the photoexcitation by the pump addresses states created during the annealing step at the ZnO surface.

The above observation compares well with previous work by Ref. ⁵¹, who found that oxygen vacancies are formed above 700 K with increasing efficiency as the temperature is raised. Hence, it seems likely that oxygen vacancies cause the deep defect levels.“

2. A probing energy of 4.25 eV is selected to access the VB information in the measurement. How did the authors decide to use this value instead of others? Could the authors provide a brief discussion in this aspect?

Indeed, this should be clarified. The choice of 4.25 eV as the probing photon energy for the VB is a consequence of two requirements: Firstly, photoemission from the VB maximum requires 7.6 eV, because the sample work function is 4.4 eV and the VB is 3.2 eV below the Fermi level E_F . This is above the highest photon energy provided by our laser system, which is 6.3 eV. Therefore, the requirement can only be met with two-photon-photoemission (2PPE) with a photon energy of at least $7.6 \text{ eV}/2 = 3.8 \text{ eV}$. Secondly, the work function is 4.4 eV. To avoid a background signal due to

one-photon-photoemission (1PPE) the probe photon energy has to be below 4.4 eV, but as high as possible in order to reduce spectral overlap of the VB feature with secondary electrons (cf Fig. 3b in the manuscript).

Related changes in the manuscript

- Added an energy level diagram in the inset of Fig. 3a.
- Added line 203: “A probe photon energy of $h\nu_{\text{probe}} = 4.25$ eV gives access to the VB in a two-photon photoemission process (see the inset in Fig 3a).”
- Added to the caption of Figure 3a: “This photon energy is below the work function of the sample ($\Phi = 4.4$ eV) and the VB is accessed by two-photon photoemission, see the energy sketch in the inset.”

3. Could the authors clarify more in details for the argument mentioned in pp. 5, line 150-153? Why the positive surface charges will decelerate the photoelectrons only at the small negative pump-probe delay?

For negative pump-probe delays the probe laser pulse arrives before the pump laser pulse and photoemits electrons. For small negative delays, the probe electrons are just emitted by the probe laser pulse at the time the pump laser pulse arrives. Hence, they are close to the surface. For large negative delays, the electrons emitted by the probe laser pulse had time to travel to the detector and are not anymore near the surface when the pump laser pulse arrives. Consequently, electrons emitted by the probe laser pulse are only decelerated by the pump-induced field at small negative pump-probe delays. For an in-depth explanation of the negative delay dynamics, please see ref. [Yang, S. L., Sobota, J. A., Kirchmann, P. S. & Shen, Z. X. Appl. Phys. A **116**, 85–90 (2014), DOI 10.1007/s00339-013-8154-9].

Related changes in the manuscript:

- The paragraph about the negative delay dynamics was rephrased:
Line 217: “Beyond monitoring band positions, trARPES offers another direct way of probing that the surface is indeed positively charged after photoexcitation: As shown previously for GaAs⁴⁷, a change of the surface charge leads to a short-range electrostatic field that extends into the near-surface vacuum region. In a pump-probe experiment, photoelectrons emitted by the probe laser pulse are decelerated by this pump-induced field at small *negative* pump-probe delays $\Delta t < 0$, when the photoelectrons are still close to the surface⁴⁷. As discussed in great detail by Yang et al.⁴⁷, this effect leads to a downward shift of the energetic position of the detected probe electrons as a function of negative pump-probe delay. The shift is

strongest close to time zero and gets weaker at larger negative pump-probe delays^{47,48}. In our experiment, the data in Fig. 1d clearly shows that such a downward shift of the photostationary state signal probed by $h\nu_{\text{probe}} = 6.3$ eV occurs for $\Delta t < 0$ on a 100 ps timescale. The downward shift at negative delays unambiguously demonstrates that the surface is positively charged due to photoexcitation by the pump laser pulse.”

4. It is evident that the deep defect states are responsible for this transition process; however, the origin of these defects are not mentioned in the manuscript. It will be better if the authors can discuss more about the physical origin of the deep defect states. This could help the readers find possible engineering methods to demonstrate similar phenomenon in other types of transition metal oxide (TMO) materials.

We have included additional experimental evidence showing that the deep defect states are produced during the sample preparation procedure by high temperature treatment and concluded that the most likely defect is the oxygen vacancy (see our answer to question 1)). Independent of the chemical nature of the defects, our results show that they must form a donor-type in-gap state. The in-gap state must be deep enough in the band gap to be neutral in thermodynamic equilibrium, i.e. not spontaneously ionized due to thermal activation. Acceptor-type defects such as Zinc vacancies can be excluded as we write in the revised manuscript:

Line 263: “However, zinc vacancies can be ruled out as they should be negatively charged in thermodynamic equilibrium²⁹. Photoexcitation of such states would therefore only diminish the number of negative charges at the surface, and hence *reduce existing* upward BB instead of *creating* downward BB. In contrast, oxygen vacancies are neutral in the equilibrium and induce a state 0.4 eV above the VB maximum [...].”

Related changes in the manuscript

- In the abstract and conclusions, added “donor-type” to the expression “deep surface defects.”

5. In the beginning of this manuscript, it is found that there is lack of discussion in terms of the potential applications for this ultra-fast and low fluence SMT in ZnO.

We agree and now explicitly discuss such applications in the introduction:

Related changes in the manuscript

- Line 61: “For ZnO, a photoinduced, *ultrafast* control of the conduction properties would be especially appealing, as any application would benefit from the ease of nanostructuring and transparency to visible light of this material^{29,30}. At semiconductor *surfaces*, the photoexcitation of defects would imitate the effect of the gate terminal in field effect transistors. Such phototransistors could then be used for the control of ultrafast currents in information technology or optoelectronic devices, such as light emitters in the terahertz regime³¹.”

6. In line 154, the electron signal was downward shifted in the negative delay region. It was explained as a photoelectron deacceleration by the pump-induced positive surface charges. What do the black dots in Fig.1(a) stand for? And why there is no similar shift in negative delay region in VB as shown in Fig.3(a)?

We agree with the referee that the origin of the photoelectron signal at negative pump-probe delays should be discussed. We now expanded Fig. 1 and added substantially more information in the main text. This state is attributed to long-lived defect excitons forming a photostationary state in our system (line 89 to 118 in the revised manuscript). The black dots is the position of the peak maximum of the photoelectron intensity distribution of this state.

Related changes in the manuscript

- Added two panels to Fig. 1 with spectra at negative pump-probe delays/with the pump laser pulse off.
- Added to Figure caption [...] “Black dots: Peak maximum of the intensity distribution.”
- Lines 89 to 118: Additional section about the origin of the signal detected by the probe at negative delays.

We now turn to the question why a shift at negative pump-probe delays is not apparent in case of the VB in Fig. 3a. A shift of the valence band at negative delays *does* occur, but it is not discernable in Fig. 3a. The shift of the VB spectrum is shown in Figure R3 a): VB spectra (probed by 2PPE with 4.25 eV) are displayed for two different negative pump-probe delays, -100 ps and -1 ps. Clearly, the VB edge is shifted downward at small negative versus large negative delays. The delay dependency of the shift is shown in Figure R3 b). The solid lines are fits with a model that describes the short-ranged pump-induced field. Note that the VB shift is smaller than that of the defect exciton state within the accessed time window. This is because the VB electrons are slower than those that arise from the defect excitons. Low velocity electron experience the pump-induced field affects them at larger pump-probe delays (up to about 300 ps, cf. Fig. R3 b). This means that the shift is stretched over a

larger delay window. In addition, because the surface metal and hence the pump-induced surface dipole density decays within $\tau_3=213(19)$ ps, the pump-induced field partially decays within this time window. This is expected to reduce the observed shift [Tanaka, *J. Electron Spectros. Relat. Phenomena* **185**, 152–158 (2012), <http://dx.doi.org/10.1016/j.elspec.2012.06.003>]. The evaluation of the negative delay dynamics goes beyond the scope of the present manuscript and will be published separately. For referee information, we append the chapter of the doctoral thesis of Lukas Gierster, which will be submitted in the coming weeks containing this evaluation and an extended discussion.

Figure R3 Pump-induced shifts on the VB spectrum at negative pump-probe delays. a) a) VB spectra probed by $h\nu_{\text{Probe}}=4.26$ eV at -1 ps and -100 ps ($h\nu_{\text{Pump}}=3.43$ eV). The VB is shifted down close to time zero with respect to large negative delays. b) Extracted shift as a function of pump-probe delay for two different pump fluences, $25 \mu\text{J}/\text{cm}^2$ and $50 \mu\text{J}/\text{cm}^2$, relative to the value at large negative delays.

7. In line 167 the author introduced a method to address the in-gap states. It was achieved via tuning the pump energy ($h\nu_{\text{pump}}=3.0/3.2$ eV) and recording the threshold energy that triggered the PE intensity shift. However, there was no evidence that the photon excitation was purely resulted from deep-level trap delocalization.

Apparently, the corresponding part of the manuscript wasn't phrased clear enough. In the new version of the manuscript we have clarified that below 3.4 eV no other excitations in the system than the depopulation of deep defects are possible:

Line 249: "Fig. 3d shows that downward BB is still induced by photoexcitation with $h\nu_{\text{pump}} = 3.2$ eV (full width half maximum: 0.1 eV). This unambiguously confirms that, downward BB, and hence the PIPT, is not driven by excitations across the ZnO band gap (3.4 eV)."

Because the valence band shift can be induced with below band gap excitation and the shift evolves concomitant with the electron delocalization in the ultrafast time domain as shown in Figure 2a, we

conclude that the photodepletion of deep defects is at the origin of the photoinduced semiconductor-to-metal transition.

8. In line 192 the influence of temperature was mentioned (100K-256K). It was depicted in Extended Data Fig. 3 that higher PE intensity was observed at higher temperature. What I observe is that at 256K, the equilibrium PE intensity is almost the same as the excited one ($\Delta t=50\text{fs}$) at 97K. Does it imply that the PE intensity variation could also be a result of heating by the laser?

If we understand the comment correctly, the referee interprets the data in Extended Data Fig. 3 as a comparison of the equilibrium PE intensity at elevated temperature with the excited one at low temperature. This is not the case. The figure shows the *photoinduced* photoelectron intensity probed by 6.3 eV at $\Delta t=50\text{fs}$ for three different sample temperatures. Clearly, the photoinduced metallic intensities are similar for all base temperatures. We conclude that the metallization does not require cryogenic temperatures. We have modified the Figure caption to avoid misunderstandings.

A thermal origin of the photoinduced semiconductor-to-metal transition can be excluded due to the fast time scale of 20(20) fs at which the metallization occurs. Moreover, the phase transition can not be thermally induced because there is no metallic phase in the equilibrium phase diagram of ZnO as a function of temperature. [Lien et al., J. Appl. Phys. 110, 063706 (2011); <https://doi.org/10.1063/1.3638120>]

9. Overall, the manuscript provides a detailed description on the PIPT method to monitor the energy states at the ZnO surface. It claims that the SMT after photoexcitation is driven by BB. The method is interesting to me, but the result is “not surprising”. In the view of providing stronger evidence and enriching the manuscript, I suggest adding more probing methods or compare the PIPT results with other methods. For instance, DLTS can be used as a supporting data to the argument of the deep level traps.

The referee states that the result is “not surprising”. Indeed, as outlined in the introduction of our manuscript, several aspects of our results could probably have been expected based on previous, mostly static experiments and theory. As a matter of fact, despite the simple elegance of the underlying physics, no-one has ever predicted, nor observed the phenomena that we present. There is no demonstration of such a low-fluence PIPT in the literature, no PIPT in ZnO, no PIPT of this kind at a semiconductor surface, no study with such fast back relaxation, no exciton Mott transition at the

equilibrium E_F and no real-time observation of dynamic delocalization and localization. Finally, no critical behavior of effective masses in the ultrafast time domain has ever been observed - although one could have guessed that all this could happen. From our point of view, this is the beauty of this study: A strong and clear effect, in a large set of observables that does not require the involvement of strong correlation effects, but can be elegantly explained by textbook semiconductor physics – applied to ultrafast, highly non-equilibrium dynamics.

In order to understand photoinduced phase transitions, usually complementary techniques are required. Yet, we believe that, in the present case, trARPES alone deciphers the mechanism *en detail*, not least because the experiments provide so many different observables and the many systematics that we performed. In this way, the elementary processes occurring upon photoexcitation are not proposed based on one set of data, but confirmed by – sometimes even several – test experiments. For the convenience of the referee, we summarize this below:

1. The evolution of the effective mass with photoexcitation fluence. We have added additional evaluation in the revised manuscript that shows that this is precisely what is expected from the Mott-Hubbard theory of the Mott transition between shallow donor dopants and what is observed upon chemical doping in 3D and 2D.
2. Transient downward band bending is evidenced by measuring the valence band position in the ultrafast time domain. The valence band position evolves in exact correlation with the occupation and effective mass of the metallic band in the ultrafast time domain. The correlation between the metallization and the downward shift of the valence band is precisely what is observed upon chemical doping of semiconductor surfaces.
3. Positive surface charging is evidenced from the negative delay dynamics – an experimental corroboration that is *unique* to the trARPES technique.
4. The downward band bending, and hence the surface metallization, can be driven by below band gap excitation. This evidences that the photoexcitation of deep defects in the ZnO band gap is at the origin of the PIPT.
5. The annealing temperature dependence (Fig. 4 in the revised manuscript) confirms that the photoresponse depends on the defect density and suggests that these defects are oxygen vacancies.

We believe that our data provides sufficient evidence for the proposed scenario and hope that it will trigger numerous follow-up works that provide complementary information on the observed phenomena.

10. Most of the fonts are not very clear. Please improve the graph.

Apologies, this likely was a result of the article compression upon upload. In the current submission, we upload high quality figures separately.

Reply to Referee 3

The manuscript by L. Gierster et al. reports on the experimental observation of the ultrafast generation and decay of a metallic surface state in ZnO upon weak photoexcitation. The reported results are without doubt novel and very interesting and envision potential future applications particularly with respect to the applied fluence range as well as the observed characteristic (ultrashort) time scales. The presented trARPES data are of very high quality and in general support the interpretation of the authors (see, however, my comment below). Particularly the formation of a (real) transient metallic state on extremely short timescales is convincingly proven by the experiment. Based on the topic, the novelty of the presented results and the convincing data the manuscript seems to me very suitable for publication in Nature Communications. However, before a final decision I would like to ask the authors to consider the following comments:

The authors argue that the formation of the metallic state is result of a (rigid?) shift of the conduction band below the Fermi level upon photoexcitation. This is implied by Figure 1(a) and also suggested as the favorable scenario later in the manuscript. However, in the transient data [Figure 1(b) and also the attached movie] I cannot see indications for such a transient band shift (a shift of the peak maximum associated with the band) neither during the formation nor during the decay of the metallic state. Instead, the authors convincingly show that the effective mass of the band is changing as a function of time, excluding at least a rigid(!) shift of the conduction band. As an alternative I am wondering, whether (also) the following scenario could apply: As the excited carrier density is increased during the absorption of the pump pulse, first, an excitonic state right at the Fermi level is formed that then transiently undergoes an exciton Mott-transition. Such a scenario could explain the change in the effective mass as a function of time and seems also to be supported by the fluence dependent data shown in Figure 2. I would like to emphasize that such an interpretation would not contradict the key message of the paper. However, it would rather fit to the first scenario listed in the introductory part of the manuscript.

We thank the referee for his/her positive assessment regarding our results and the acknowledgement of our data quality as well as of the novelty of our results.

The first comment makes an excellent point. Indeed, the photoinduced phase transition cannot be grasped within a picture involving rigid band shifts. This was already suggested in the original manuscript, but not discussed in detail. The referee is correct – a rigid shift of the conduction band toward the surface does not occur and the PIPT needs to be considered as a Mott transition between

excitonic states. It should be noted, though, that it is a Mott transition between *defect* excitons and not between *free* excitons as now discussed in detail in the manuscript. By defect excitons we mean an electron that is bound to a photohole at a defect site. As the deep defects are located at the surface, this creates a surface potential change. This surface potential change causes band bending – locally. We realized that the localized character of the band bending (for low densities) did not become sufficiently clear in the previous version and have emphasized this aspect in the new manuscript.

This significant re-shaping of the discussion can be found throughout the manuscript and we kindly ask the referee to refer to our paper for further details. We appreciate the referee’s comment very much, as it helped to improve the manuscript significantly.

Additional related changes to the manuscript:

- Modified Fig 1a (now Fig 1c). The sketch now shows the metallic band formation between defect excitons instead of suggesting a rigid band shift.
- Modified the abstract to incorporate the view of a Mott transition between surface confined defect excitons.
- Added the local aspect of the downward band bending in the introduction. In the first paragraph of the introduction we write “At low doping density, the BB is concentrated around isolated electron pockets at the surface^{8,9}. Electron delocalization occurs above a critical electron density only².”
- In order to distinguish the first and third scenario listed in the introduction we write now:
 - “1. At a critical excitation fluence a Mott transition between free excitons occurs, leading to the formation of an electron-hole plasma with quasi-Fermi levels in the CB and VB^{11,12}. This SMT is closely related to the above described Mott-Anderson transition¹³. [...] however, without a change of the equilibrium band structure, there is no density of states around the equilibrium Fermi level E_F and no true SMT has occurred.
 2. [...]
 3. Analogous to chemical doping, the band structure can also be optically manipulated by photodoping^{22–24}. One pathway is the photoexcitation of deep donors, which creates electron-hole pairs bound in hydrogenic potentials, with the hole localized at the impurity site [...]. In contrast to *free* excitons, these *defect* excitons are fixed in space, forming a direct analogue to the shallow donors discussed above²⁵. As for chemical dopants, the geometric confinement of the photoholes to the surface modifies the surface potential, thereby causing

local downward BB. At a critical density of such states, an impurity band forms (Fig. 1c) and the surface is metallized.”

- Motivated by the referee’s comment, we added an additional paragraph in the manuscript (line 168 to 178) and the fit in Fig. 2b to discuss the effective mass change as a function of pump laser fluence. The effective mass shows critical behavior, as predicted from the Mott-Hubbard theory of the Mott SMT.
- Rephrased lines 128 to 136 of the original manuscript. Now the paragraph explicitly discusses why a Mott transition between free excitons *without changes to the equilibrium electronic band structure* can not account for the experimental findings. Instead, a Mott transition between surface-confined defect excitons is the proposed scenario:

Line 187: “A crucial aspect of the photoinduced metal phase is that the quasi- E_F in the metallic band equals the equilibrium E_F for all excitation densities (cf. Fig. 1f and extended data Fig. 2). This means that the SMT goes beyond a Mott transition between free excitons without changes to the equilibrium electronic band structure: The exciton binding energy in ZnO is 60 meV, which is not enough to create a state at the Fermi level, as the equilibrium E_F is located 200 meV below the bulk CB. Thus, at low fluence, no electronic states below E_F due to free excitons can arise. Likewise, above the Mott density, the free electron-like population in the CB would be 200 meV above E_F . Photoexcitation must therefore change the equilibrium band structure. This could be reached via band gap renormalization (BGR) due to carrier-carrier screening¹¹. BGR would shift the CB downward, and at the same time the VB would move *upward*⁴⁶. However, also by photodoping with defect excitons, electron energies below those expected for free excitons could be reached if confined to the surface: The depopulation of surface defects would result in positive surface charges. This modification of the surface potential leads to local downward BB toward the surface, as upon chemical doping of semiconductor surfaces. The *local* downward BB can reach several hundred meV below the Mott limit, as is known for ZnO doped with hydrogen^{7,8}. In this case, contrary to BGR, the VB should shift *downward*.” [Continues with showing that the VB shifts downward.]

- Modified the discussion (lines 279 to 303) with regard to the Mott transition between defect excitons.

In Fig.1(b) before time-zero a spectral feature of minimum PE intensity is marked which shows a transient behavior due to the interaction with the positively charged surface (this detail is discussed in the manuscript). Can the authors comment on the origin of this spectral feature: In the movie one recognizes also for this feature a parabolic dispersion out of which the metallic band is kind of emerging.

We mentioned the nature of the signal at negative delays only briefly in the previous version. The referee's comment, combined with the curiosity of all referees with regard to the photostationary signal and the response at negative delays, has convinced us to provide a more in-depth discussion of the matter.

We now show an angle resolved spectrum at negative delays in the manuscript (additional panel in Fig. 1) with an evaluation of the apparent dispersion, which is flat, characteristic for localized states. In an additional section we explain the origin of this feature: A fraction of the pump-induced defect exciton population shows lifetimes exceeding the inverse repetition rate of our laser system (5 μ s). This long-lived population shows up at negative pump-probe delays.

Thus, prior to the transition to the metallic state, the sample is already doped with shallow donor-like defect excitons, but below the critical density required to form a metallic band.

Related changes in the manuscript

- Added two panels to Figure 1 with photoelectron spectra (angle-integrated and angle-resolved) at negative delays/with the pump off.
- Added Line 89 to 118: Section "Photostationary n-type doping."

REVIEWERS' COMMENTS

Reviewer #1 (Remarks to the Author):

The authors have made extensive discussions in the response to all three referees' comments, and have addressed my earlier concerns. I think the revised manuscript is now much improved. I would recommend it for publication in Nature Communications, and just have the following suggestion.

The newly added Figure 4 shows more results to support the argument of the role of surface defect in the photoinduced transition. It would be nice to present the photoemission spectra in a wider range covering the valence band and defect state. This will first provide a more direct/clear comparison between those two annealing conditions, as the data is in arbitrary unit. Moreover, one may also expect to see a stronger defect state intensity and more photoinduced depletion in the case of 950 K annealing, although it is in the vicinity of the valence band.

Reviewer #2 (Remarks to the Author):

The authors have addressed the questions satisfactorily and the revised manuscript has been improved. On this basis, I recommend its publication in Nature Communications.

Reviewer #3 (Remarks to the Author):

I very much appreciate the very detailed reply of the authors to the reports of all reviewers. In my opinion, the concerns of the reviewers have been addressed in the reply as well as in the revised version of the manuscript in a very comprehensive and convincing manner. In conclusion, I strongly recommend publication of the manuscript in Nature Communications.

Reply to Referee 1

The authors have made extensive discussions in the response to all three referees' comments, and have addressed my earlier concerns. I think the revised manuscript is now much improved. I would recommend it for publication in Nature Communications, and just have the following suggestion.

The newly added Figure 4 shows more results to support the argument of the role of surface defect in the photoinduced transition. It would be nice to present the photoemission spectra in a wider range covering the valence band and defect state. This will first provide a more direct/clear comparison between those two annealing conditions, as the data is in arbitrary unit. Moreover, one may also expect to see a stronger defect state intensity and more photoinduced depletion in the case of 950 K annealing, although it is in the vicinity of the valence band.

We thank the referee for reading our revised manuscript carefully and are pleased to hear that his/her earlier concerns have been addressed. The suggestion above is a good point, but we cannot extend the energy axis to such low energies, as explained in the following.

The data in Figure 4 is recorded with a probe photon energy of $h\nu_{\text{Probe}}=6.3$ eV. Because the sample work function is 4.4 eV, electrons can be photoemitted from electronic states with binding energies up to 1.9 eV with respect to the Fermi level E_F . As illustrated in Figure R1, with the probe photon energy $h\nu_{\text{Probe}}=6.3$ eV it is not possible to access the valence band and defect states, which have binding energies >3.0 eV. This is also why we used two 4.25 eV photons in order to monitor the dynamics of the VB.

Therefore, extending the energy axis in Fig. 4 up to the range covering the valence band and defect states is not possible. However, although the shown photoelectron intensity is in arbitrary units, it is still proportional to the electron density of the pump-induced metal phase. The spectra for the two different annealing temperatures (750 K, 950 K) were recorded with the exact same experimental conditions, i.e. on the same day with the same pump and probe laser fluence, respectively. Hence it is possible to compare them. We did not explicitly mention this in the previous version of the manuscript, but do so now:

Related changes in the manuscript

Caption of Fig. 4, added: *"The same fluence of pump as well as probe laser pulses is used for both shown spectra, hence the absolute PE-intensity can be compared."*

Figure R1 Energy level diagram to illustrate the accessed energy region for photoelectron spectroscopy with $h\nu_{\text{Probe}} = 6.3 \text{ eV}$. Electrons from the defect states and the valence band maximum (VBM) cannot be promoted above the vacuum level E_{vac} with this photon energy and hence are not photoemitted.